# When Does the Prey/Predator Ratio Work for the Effective Biocontrol of Cotton Aphid on Cotton Seedlings?

**DOI:** 10.3390/insects13050400

**Published:** 2022-04-21

**Authors:** Ping Zhang, Yao Lu, Wendi Chao, Zhaoke Dong, Abid Ali, Tong-Xian Liu, Zhaozhi Lu

**Affiliations:** 1Shandong Engineering Research Center for Environmentally Friendly Agricultural Pest Management, College of Plant Health and Medicine, Qingdao Agriculture University, Qingdao 266109, China; zpsds2018@163.com (P.Z.); lydhr456@163.com (Y.L.); chaowendi@163.com (W.C.); zhaoke_dong@126.com (Z.D.); 2State Key Laboratory of Desert and Oasis Ecology, Xinjiang Institute of Ecology and Geography, Chinese Academy of Sciences, Urumqi 830011, China; 3The Specimen Museum of Xinjiang Institute of Ecology and Geography, Chinese Academy of Sciences, Urumqi 830011, China; 4University of Chinese Academy of Sciences, Beijing 100049, China; 5Pherobio Technology Co., Ltd., Beijing 101100, China; 6Department of Entomology, University of Agriculture, Faisalabad 38040, Pakistan; abid_ento74@yahoo.com; 7College of Life Science, Shenyang Normal University, Shenyang 110034, China; 8Center for Science and Technology Diplomacy Research, School of Politics and International Relations, Lanzhou University, Lanzhou 730000, China

**Keywords:** *Coccinella septempunctata*, critical threshold, effectiveness of biocontrol, population growth rate

## Abstract

**Simple Summary:**

Cotton aphid (*Aphis gossypii* Glover) (Homoptera: Aphididae) is a major pest of cotton and other cash crops across cotton-growing areas in China. The decision to spray insecticide or rely on biocontrol and delay spraying is often a dilemma in cotton-growing regions. In this study, we used a laboratory experiment and a caged experiment in a garden as well as modeling to understand the ratio of prey/predator for the abovementioned decision making. We suggested that the ratio of prey/predator should be less than 450 for the effective biocontrol of cotton aphid at cotton seedling stage. This finding can improve the efficiency of larger-scale cotton aphid management in China.

**Abstract:**

The decision to delay or cancel spraying insecticides against pest aphids is dependent on the ratio of prey/predator, which reflects how well the predator can suppress the aphid population increase in the field. It is challenging to estimate the ratio of prey/predator due to the multiple factors involved in the interaction between prey and predator. Cotton aphid (*Aphis gossypii* Glover) is a serious pest, widely distributed in cotton-growing areas around the world. We combined different ratios of aphids with aphid oligophagous ladybird beetles (*Coccinella septempunctata* Linnaeus) under laboratory and garden conditions to investigate the critical threshold for prey/predator which effectively reduced the cotton aphid population increase. Two kinds of modeling were developed to understand the relationships between the ratio of prey/predator and the PGR (population growth rate), and with the effectiveness of biocontrol (EBC). We found the critical values of PGR should be less than −0.0806 (predators artificially released after 5 days) and then less than −0.075 (predators released after 10 days) if EBC is less than 50%. We recommend that the ratio of prey/predator should be less than 450 for the effective biocontrol of cotton aphids at the cotton seedling stage. These values can be reference indices for the management of aphids in mid-summer.

## 1. Introduction

Biocontrol services, including natural enemies, are sustainable tools for managing pests whilst conserving the environment, biodiversity, and food production [1,2]. In agricultural production, the contribution of biocontrol to increasing yield and crop quality was relatively small (<20%) [3], but its value to the economy and wellbeing of society is crucial and underestimated [4,5,6]. For example, natural suppression of soybean aphid in soybean was worth an average of USD 33 ha^−1^ [7]. Numerous technologies promoting the efficacy of biocontrol, such as selective insecticides [8,9], environmentally friendly bio-pesticides [10,11], augmentative biological control [12,13], habitat management [14,15], and genetically engineered crops [16,17,18], have been implemented in agricultural management. Conservation biological control (CBC), which integrates natural enemies back in to crop systems, can minimize the likelihood of a pest outbreak, and reduce pesticide use and pest resurgence [8,19]. Additionally, CBC is more practical, economical, environmentally friendly, and more feasible for manipulation in an open field [20,21]. Therefore, CBC has been used effectively in multiple ecosystems throughout the world [19,22].

Pesticides may be used in CBC to assist in reducing higher densities of pests when there is a deficit of biocontrol agents in some cases. The threshold for insecticide application is important to avoid unnecessary spraying, which would also kill natural enemies. There are two factors, the economic threshold level (ETL) and the ratio of natural enemies to prey, that determine the threshold. The prey/predator ratio has been employed successfully to decide the releasing frequency and quantity of predators with various pests, particularly in microcosm ecosystems [23,24,25,26,27], but it has been used less in open fields [28] due to the complex interactions between prey and predator under the influence of multiple factors in nature. It is necessary to compromise between pesticide use patterns and the biocontrol service of natural enemies when the pest density is near to the ETL, and we need to build some useful parameters such as the ratio of prey/predator to solve this issue. There is still a crucial bottleneck for the effectiveness of CBC in various cropping systems.

Cotton aphid (*Aphis gossypii* Glover) is one of the serious r-strategy pests in cotton-growing areas worldwide, causing economic losses from 10% to 30% [29,30,31]. Populations have persisted within a cotton growing season for 4–5 months in various geographic areas [32,33,34], where it is an early-through mid-season pest: extending to late season if broad-spectrum insecticides have reduced the role of natural enemies [33]. Natural enemies can regulate aphid populations effectively and mitigate outbreaks and yield loss [17,30,33,34,35,36]. Previous studies recommended that a ratio of prey/predator of less than 300 in a cotton crop at seedling stage would indicate efficient biocontrol [37,38] and insecticide application would not be required. In China, there have been higher levels of resistance in aphids to pesticides [39,40]. Consequently, a critical threshold ratio of aphid/predator needs to be quantified that will best inform decisions about appropriate insecticide spraying and the effectiveness of natural enemies, and thus help mitigate the resistance of cotton aphid to insecticides.

In this study, we conducted a laboratory experiment and a caged experiment in a garden to investigate aspects of the complex interaction between cotton aphid and a predator, the ladybird beetle, which is the dominant and most effective natural enemy in open cotton fields, and to determine with confidence the ratio of aphid/predator that should provide biocontrol of cotton aphid. The following questions were addressed: (1) How does the prey/predator ratio affect the population growth rate of aphid? (2) Does the ratio of prey to predator affect the efficacy of biocontrol for aphid?

## 2. Materials and Methods

The experiments were conducted under constant temperatures in laboratory and fluctuating temperatures in a garden in Xinjiang Institute of Ecology and Geography, Chinese Academy of Sciences, Urumqi, Xinjiang, China (43°51′57″ N, 87°33′51″ E). The average daily temperature ranged from 9 °C to 35 °C, the average temperature was 20 °C (for details see the Appendix A).

Cotton aphids were collected from Shihezi Agriculture Station, Xinjiang Province, China (44°20′2″ N, 86°02′42″ E), and transferred to cotton seedlings in the laboratory and cultured for three generations before use in the experiments. The predator ladybird beetles (*Coccinella septempunctata* Linnaeus) were purchased from the Fujian Yan Xuan Biological Technology Co. Ltd. (Fuzhou, China).

The cotton seedlings (Xinluzao 61 variety) were planted in plastic pots (length, height, and width of 20, 14, and 16 cm, respectively) with commercial potting mix, with 1 plant per pot. A total of 2400 plants were grown in an enclosed, insect-proof greenhouse of 30 m^2^ and used at the 6–7 euphylla leaf stage.

Experiment 1: Biocontrol under stable temperature regimes in the laboratory. Three factors were considered: (1) temperature at 3 levels—24 °C, 27 °C, and 30 °C; (2) predator ladybird beetles (*C. septempunctata)* at 4 levels—0, 1, 2, and 4 predators per cotton plant according to long-term field data (Lu, unpublished); and (3) aphid density (200, 400, and 800 per cotton plant). This experiment was conducted in a complete randomized design, with 36 treatments employed; each treatment had 5 simultaneous replications.

Each plant was enclosed in a nylon net (diameter: 0.24 mm) cage (length, height, and width of 20, 20, and 30 cm respectively) after transferring 25–70 adult aphids onto each plant 1 week prior. After one week, we counted the aphids on each plant and removed the surplus to the number required for each experiment (200, 400, or 800 aphids in each plant cage). Simultaneously, the beetles were added to the plant as required for each treatment. Cages were transferred into climate-controlled chambers (PXY-250Q-A, Keli experimental company, Shaoguan, China) at the required temperatures. Aphids on each plant were counted with 8 observers at 5-day intervals until their population steadily dropped near to 0. 

Experiment 2: Biocontrol under fluctuating ambient temperatures in the caged experiment in the garden (CEG). The same levels of aphid density (200, 400, and 800 per plant) and ladybird beetles (0, 1, 2, and 4 predators per plant) were used in the caged experiment in the garden as in the laboratory experiments. The plants, aphids, and ladybird beetles were managed as described in Experiment 1. As in Experiment 1, after transferring aphids and predators, each plant was enclosed in a nylon net (diameter: 0.24 mm) cage (length, height, and width of 20, 20, and 30 cm, respectively). Then, all cages with treated plants were placed in garden under natural environmental conditions (called caged experiment in the garden—CEG). A total of 12 treatments with 5 replications were conducted simultaneously. The aphid count was carried out as in Experiment 1. The weather data were obtained from the local weather bureau. 

### Statistics

The population growth rate (PGR) is defined as: $$r=(lnN_{t}-lnN_{0})/t$$
where $N_{t}$ is the number of aphids at a given time, $N_{0}$ is the initial number of aphids in each treatment, and *t* is days interval between $N_{t}$ and $N_{0}$ [41].

The effectiveness of biocontrol (EBC) is used to describe biocontrol efficacy and is calculated as follows: $$\mathrm{EBC}=(N_{ck}-N_{p})/N_{ck}\times 100\%$$
where $N_{ck}$ and $N_{p}$ are the number of aphids in the predator-absent treatment and the predator-present treatment, respectively, with the other factors the same in both treatments. A simple linear regression was used to describe the relationships between (r) and the prey/predator ratio, and between EBC and the prey/predator ratio using Origin 9.0 software (OriginLab Corporation, Northampton, MA, USA).

Because cotton aphids developed their abundance rapidly but reduced in 20–30 days, and then declined in the field, we concerned the effectiveness of biocontrol within 10 days. Therefore, the EBC in the caged experiment in the garden and the laboratory experiment under different treatments was compared at 5 days and 10 days after releasing the predator using a general linear model (GLM). In the caged experiment in the garden, aphid and predator numbers per seedling were fixed factors, and ECB was the dependent factor. A full factorial design was employed, and post hoc tests were used to examine the effects of initial number of aphids and the number of predators. Least significant difference (LSD) was used in the post hoc test to identify the differences between treatments. For EBC in the laboratory experiments, the numbers of aphids and predators and the temperature were fixed factors, and ECB was the dependent factor. Similarly, a full factorial design model was employed, and post hoc tests were used to examine the effects of the initial number of aphids, the number of predators, and temperature. LSD was used in the post hoc test to identify the differences between treatments.

## 3. Results

### 3.1. The Population Growth Rate of Cotton Aphid

The temperature, initial number of aphids, and numbers of predators combined to drive the population growth rate. Under all the temperature regimes (including the caged experiment in the garden and the laboratory experiment), the presence of one or more predators decreased the population growth rate (*r*). Under the same temperature, *r* decreased along with an increase in the number of predators present (Figure 1). Under all temperature regimes, *r* was slightly higher along with a greater initial number of aphids. Under increased temperature, *r* was lower when the other factors of initial aphid number and predator number were the same (Figure 1). In the caged experiment in the garden, in the treatments without predators present, the *r* was greater than 0 regardless of the initial number of aphids (Figure 2).

### 3.2. Biocontrol Efficiency (EBC) Affected by Multiple Factors

In the caged experiment in the garden, the number of predators and the initial number of aphids were the key factors determining the EBC. After releasing the predators, the number of aphids of each treatment had declined rapidly both at 5 days (predator—*F* = 230.50, *df* = 2, 33, *p* < 0.0001; initial number of aphids—*F* = 105.14, *df* = 2, 33, *p* < 0.0001) and 10 days (predator—*F* = 92.88, *df* = 2, 33, *p* < 0.0001; initial number of aphids—*F* = 94.14, *df* = 2, 33, *p* < 0.0001). The effectiveness of biocontrol (EBC) was greater than 80% except for one treatment: the EBC of the treatment with the combination of an initial number of aphids of 800 and 1 released predator was significantly lower than that of other treatments (Figure 3).

In the laboratory experiment, temperature, initial aphid number, and predator number influenced the EBC significantly 5 days after predators were released (temperature—*F *= 31.13, *df* = 2, 108, *p* < 0.0001; predator—*F* = 1726.88, *df* = 2, 108, *p* < 0.0001; initial number of cotton aphids—*F* = 1019.52, *df* = 2, 108, *p* < 0.0001) and at 10 days after predator release (temperature—*F* = 282.39, *df* = 2, 108, *p* < 0.0001; predator—*F* = 2514.91, *df* = 2, 108, *p* < 0.0001; initial number of cotton aphids—*F* = 2612.82, *df* = 2, 108, *p* < 0.0001). The EBC is less than 30% after 5 days when one predator was released and the initial number of aphids was 800 at all temperatures. The EBC is slightly different in all treatments after 5 days compared with after 10 days. In the treatment at 24 °C with an initial 800 aphids, the population of cotton aphids slightly increased after 10 days (Figure 4).

### 3.3. The Biocontrol Efficiency in Laboratory and Garden

The variation in EBC was smaller when the prey/predator ratio was less than 200 under different temperature regimes and with different initial numbers of cotton aphids. However, the variation in EBC increased when the prey/predator ratio was more than 400 (Figure 5). However, there was a negative relationship between EBC (*y*) and the prey/predator ratio (*x*) (at 5 days—y=107.667−0.082x, *R*^2^ = 0.871, *p* < 0.001; at 10 days—y=111.157−0.084x, *R*^2^ = 0.74, *p* < 0.001). With an increasing prey/predator ratio, the EBC of ladybird beetles against cotton aphids declined simultaneously. We estimated that a prey/predator ratio of 700 at 5 days after release of predators, or 728 at 10 days after release, can lead to an effective decline in EBC (<50% in both periods, usually regarded as successful control).

There was a positive relationship between PGR (*y*) and the prey/predator ratio (lnx) at both 5 and 10 days after release of the predators (5 days—y=−1.928+0.282(lnx),
*R*^2^ = 0.60, *p* < 0.001; 10 days—y=−1.477+0.214(lnx), *R*^2^ = 0.366, *p* < 0.001). When the ratio is less than 932, the PGR is less than 0 at 5 days after the predators release. Furthermore, when the ratio is less than 994, the PGR is less than 0 at 10 days after the predators release. We considered that an EBC of less than 50% as effectively suppressing the number of cotton aphid; therefore, based on the equations in Figure 6, the critical points of PGR should be less than −0.0806 at 5 days after predator release and less than −0.075 at 10 days after predator release. 

## 4. Discussion

The laboratory experiment and the caged experiment in the garden (CEG) showed that ladybird beetle predators suppressed the numbers of cotton aphids in most cases but failed when the initial number of cotton aphids was 800 with 1 predator (Figure 3 and Figure 4). This means that one predator cannot suppress the population growth of aphids under these conditions due to the greater number of offspring; this is especially the case after ten days, when the temperature was at a constant 24 °C, which is the optimal temperature for aphid growth (Figure 4). In contrast, the extreme temperature of 30 °C tended to increase the EBC in our study in treatments with a lower prey/predator ratio in the laboratory (Figure 4).

The PGR can determine the population dynamics of aphids [42]. In our study, we considered the PGR as an index to indicate the suppressing effect of the natural enemies (predator ladybird beetles). When the PGR is less than 0, the population size will decline with the passage of time. Low PGRs (negative values) indicate the rapid decline of a population. Practically, it is hard to understand the biocontrol efficiency of a natural enemy without controlled experiments in an open-field natural environment. Exclusion cages can be employed to estimate the EBC of natural enemies in the garden under fluctuating ambient temperatures, but this is a time-consuming and laborious process in larger areas. PGR is easier to calculate using the initial density, final density, and time interval (days). For the control of cotton aphids, in our study, when the PGR was less than −0.0806, the control efficiency from enemy predator exceeded 50%; this means that, if the predator is released in the period of low aphid growth, more than half of the aphid growth can be controlled. Therefore, PGR is a good reference for understanding the biocontrol service of natural enemies in the field over large areas.

In practice, the population of cotton aphid is regulated by multiple factors, such as alternating temperature, extreme temperature, and the composition of natural enemies [43]. Furthermore, other agricultural practices also regulate the population dynamics of aphids including insecticide use, plant diversity, and landscape change with the seasons in the field [44]. Therefore, the ratio of prey to predator required to provide effective biocontrol may be dynamic around our estimated value in the context of landscape changes and global warming. From our modeling, we recommend the threshold prey/predator ratio should be 728 because a ratio less than this threshold should lead to successful biocontrol of aphids (cotton aphid numbers declined up to 50% in 5–10 days compared with aphid numbers without the presence of predators). In our experiments, as well as through modeling, we merely considered the limited factors and simplified these to estimate the threshold of prey/predator in effective control of cotton aphids. These experiments and modeling should be developed as multiple parameters to deliberate the practical threshold of prey/predator against cotton aphids with ecofriendly actions.

Considering the efficacy of predators at different time intervals since release (10 days is more practical), we suggest that a critical prey/predator ratio of less than 450, which is calculated from the equation in Figure 6, should be employed for biocontrol services. If the prey/predator ratio is less than 450 in the caged experiment in the garden, spraying can be delayed or cancelled. However, monitoring should be conducted to estimate the prey/predator ratio on each day before aphid peak abundance time at cotton seedling stage, which is the critical window for aphid management. 

In China, previous studies have recommended that the ratio of prey/predator is less than 300 at cotton seedling stage [37]. Based on our experiments and modeling, when the ratio of prey/predator is less than 450, predators can suppress aphids effectively. Our finding promotes the use of biocontrol services and lessens the reliance on insecticides because of the greater threshold ratio. The cotton growing area has been shrinking in many regions in China [45], and a greater diversity of crops have been planted in the landscape, especially wheat which harbors many predators in late May. After wheat harvesting, predators can be driven to move into cotton fields [46]. In our study, we did not evaluate other ladybird beetle species in the ratio of prey/predators, such as *Harmonia axyridis* and *Propylaea japonica*, which are common in cotton fields in China [47]. Here, we suppose that the results from *C. septempunctata* can be generally applicable to the other ladybird beetle species if they were transferred as the natural enemy units [48]. Moreover, the landscape of cropping had more diversity, and offers the higher percentage of natural enemies in cropping landscapes [49,50]. Soft insecticides prevail in the agricultural production system in China [51], and the number of insecticides used has been declining rapidly with China’s regulation policies [52]. Therefore, we recommend the ratio of prey/predator of <450 as the threshold for suppressing cotton aphids effectively in China.

Using the ratio of prey/predator is a useful way forward for suppressing cotton aphids in the field, especially for aphids on cotton seedlings (spring peaking). It can help to make the critical decision on whether pesticides are sprayed or delayed in the field leading to economic savings and ecofriendly management if spraying is delayed. Moreover, the ratio of prey to predator can be used to balance aphid management between spring and summer, when the two peaks of aphid abundance occur in China [35]. Reduced or delayed spraying during the spring peak of aphids can help a greater number of predators survive, and the efficacy of this mode of biocontrol against aphids will see benefits during the summer peak in the field.

## Figures and Tables

**Figure 1 insects-13-00400-f001:**
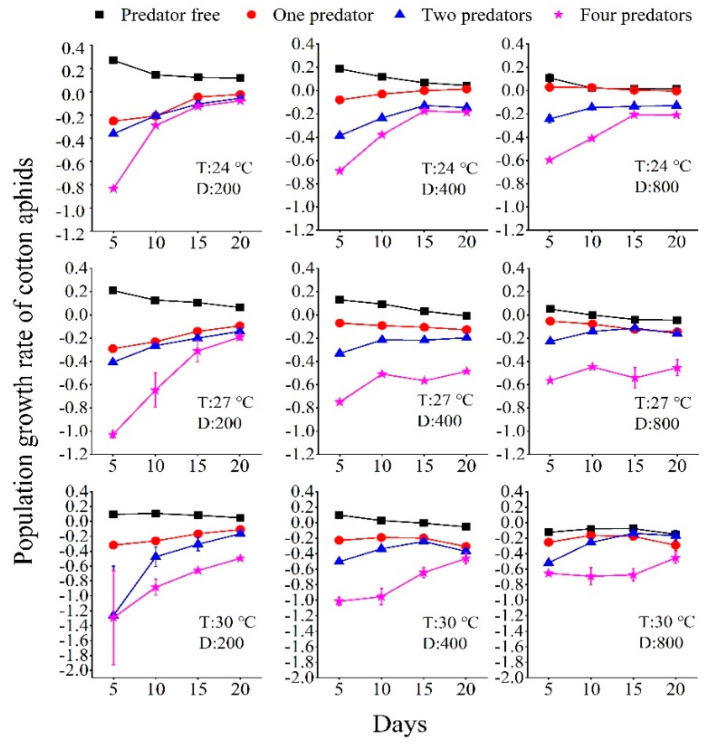
Population growth rate of cotton aphids in laboratory experiments using three initial numbers of aphids (200, 400, and 800), four numbers of predator beetles (0, 1, 2, and 4), and three constant temperature regimes (24 °C, 27 °C, and 30 °C).

**Figure 2 insects-13-00400-f002:**
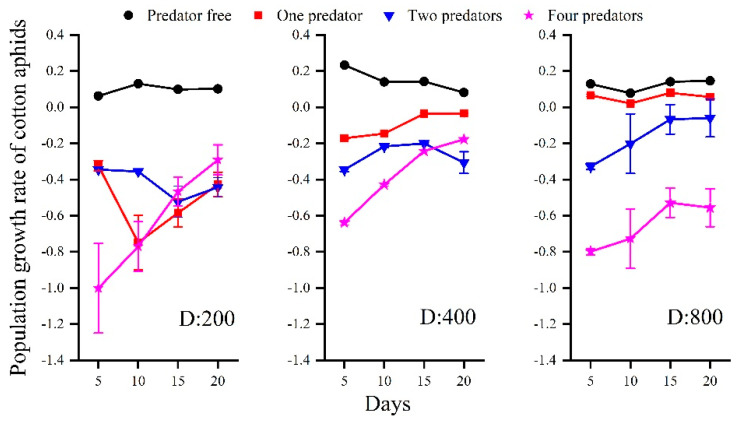
Population growth rate of cotton aphids in using three initial numbers of aphids (200, 400, and 800), and four numbers of predator beetles (0, 1, 2, and 4).

**Figure 3 insects-13-00400-f003:**
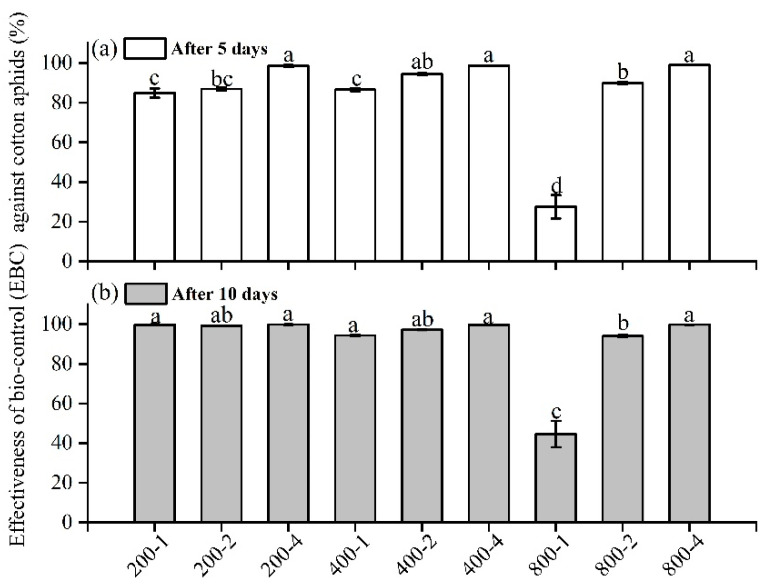
Effectiveness of biocontrol (EBC) against cotton aphid at (**a**) 5 days and (**b**) 10 days after releasing predators in the caged experiment in the garden (CEG) (the label in *x* axis presents the initial number of aphids and predators in cages). Different letters showed the mean of EBC that was different among each treatment in (**a**,**b**), while one-way ANOVA was used for comparison.

**Figure 4 insects-13-00400-f004:**
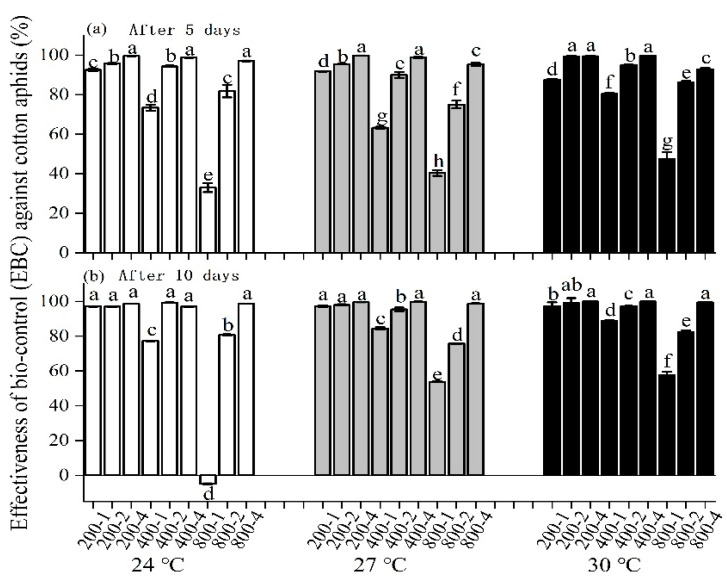
Effectiveness of biocontrol (EBC) against cotton aphids at (**a**) 5 days and (**b**) 10 days after releasing predators in laboratory experiments under various constant temperature regimes (the label in x axis presents the initial number of aphids and predators in cages).Different letters showed the mean of EBC that was different among each treatment in (**a**,**b**), while one-way ANOVA was used for comparison within one regime of temperature, respectively.

**Figure 5 insects-13-00400-f005:**
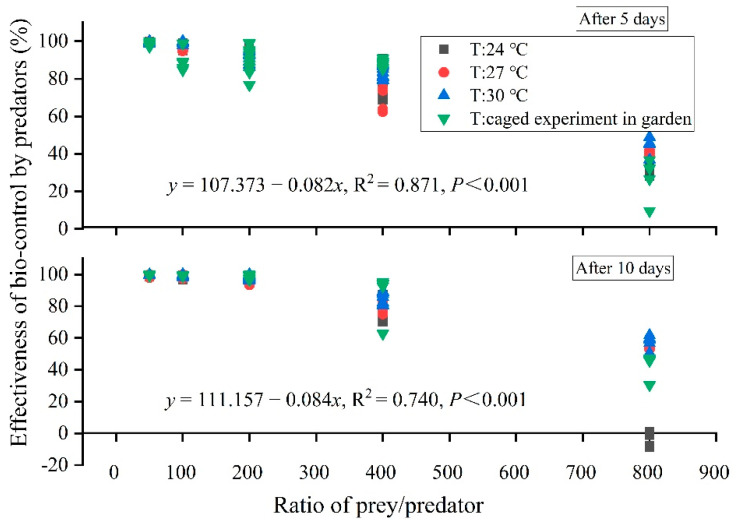
Effectiveness of biocontrol (EBC) by ladybird beetle predators against cotton aphids in relation to the prey/predator ratio in laboratory experiment and caged experiment in the garden (CEG).

**Figure 6 insects-13-00400-f006:**
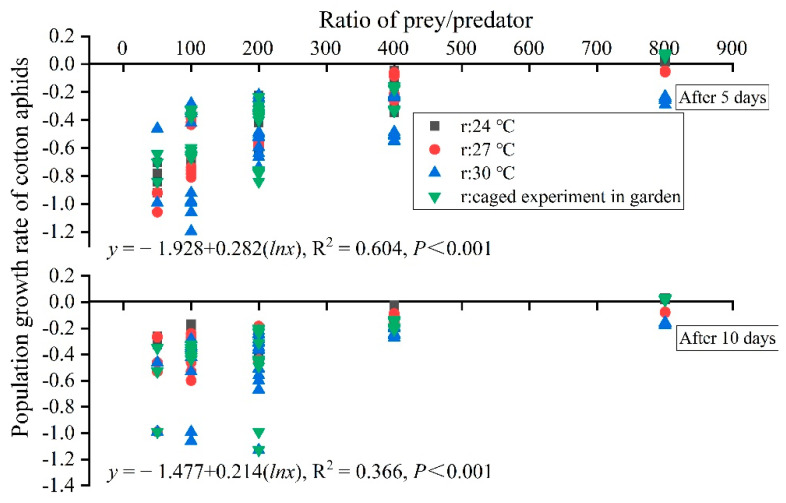
The relationship between the population growth rate (PGR as *y*) and the prey/predator ratio (lnx) in laboratory and open caged experiment in the garden (CEG).

## Data Availability

Data used in this study are available from the corresponding authors upon reasonable request.

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
