# Peer review of "When Does the Prey/Predator Ratio Work for the Effective Biocontrol of Cotton Aphid on Cotton Seedlings?"

_insects, 2022, doi:10.3390/insects13050400_

Round 1
Reviewer 1 Report
The manuscript address very important question of biological control of cotton aphid. Authors provided new data on pray/predator ratio as a measure for decision making in insecticides application. Although, it is interesting and important research I have some concerns about methodology used. Actually I suppose that methodology is appropriate but the authors did not described it adequately. My biggest concerns are about methodology used in Experiment 2. This part of the MS requires much detailed explained methodology, because most of the conclusion were drown from this experiment. Also there are parts of the Results section which must be clarified especially those related with population growth rate of cotton aphid. Based on all this I suggested major revision. All my specific comments are in additional file.
Best regards

Author Response
Dear reviewer:
Thanks very much for taking your time to review this manuscript. We really appreciate all your professional comments and suggestions. Those comments are all valuable and very helpful for revising and improving our paper, as well as the important guiding significance to our researches. We have studied comments carefully and have made correction which we hope meet with approval. Revised portion are marked in green in the manuscript. Please find our revisions in the re-submitted files. The main corrections in the paper and the responds to your comments are as flowing:
Point 1: Replace “bio-control” with “biocontrol” throughout manuscript.
Response 1: Thank for your comments. “bio-control” is replaced with “biocontrol” throughout manuscript.
Point 2: predator not predatory
Response 2: Done. We corrected all “predatory” throughout manuscript.
Point 3: Replace “natural predators or parasitoids” with “natural enemies”.
Response 3: “Natural predators or parasitoids” have been replaced with “natural enemies” (Line 60).
Point 4: Also replace “predators” with “natural enemies” because there are also parasitoids and pathogens that can be used.
Response 4: Done. The “predators” have been replaced with “natural enemies” (Line 61).
Point 5: [[33-35]: remove underline and one bracket
Response 5: Removed (Line 72).
Point 6: reorder 9-35℃ rather than 35-9℃
Response 6: Done (Line 94).
Point 7: Please provide more details. Did you use sentinel plants or plants which were planted in the field ...?Did you use isolation cages? If not how you controlled infestation from surrounding areas and plants
Response 7: Thank for your valuable comments. We are sorry to confuse you with our brief description in Experiment 2. In fact, all plants were managed as described in Experiment 1. Before the conduct of Experiment No.1, all plants were planted and sustained in an enclosed insect-proof greenhouse. After experiment, each cotton plant was transplanted in one pot in this experiment. After transferring aphids and predators, each plant was enclosed in a nylon net (diameter, 0.24 mm) cage (length-height-width: 20-20-30cm3), which avoids infection by other pests in the surrounding environment. Therefore, there was no infection from other plants.
Moreover, all cages with plants were arranged in outdoor open areas, not in the real field. It is some patchy lands in our institute. According to your suggestion, it may not be appropriate to describe Experiment 2 as a field experiment. We considered that the experiment should be described as cage experiment in garden (CEG). We have described details on line 126-132 in Experiment No. 2(CEG) and used throughout manuscript.
Point 8: Delete . It is doubled.
Response 8: Deleted (Line 134).
Point 9: “The Effectiveness of Bio-Control”, why B and C in capital?
Response 9: Revised as “The effectiveness of biocontrol” (Line 137).
Point 10: Define what is LSD. Is it least significant difference?
Response 10: Yes, it is the least significant difference (Line 152).
Point 11: I am confused with figure 1 and figure 2. More explanations is needed in Material and methods or here in results. Why those dates are chosen? If it is not important there are no need to put dates, just put 5days, 10 days, 25 days. Is 12-Sep beginning of the experiment or fifth day? If it is beginning then it is impressible to have population growth rate. Please explain figure in details. Please remove images of ladybird from the first graph because it is redundant with legend (you can use it for graphical abstract). Please make all lines uniform especially between points. What is presented with vertical bars? which appear sporadically on ranpoints.
Response 11: According to the reviewer 1 comments at No. 7 and No. 4 from reviewer 2. The dates were chosen simply to show the number aphids observed at intervals of five days. So, as you said dates was not important for observing the aphid population growth. We agreed with your suggestion to use 5, 10, 15 and 20 days instead of dates. In addition, 12-Sep is the fifth day rather than the beginning of the experiment.
Based on your comments, we revised the figure 1 and figure 2, where images of ladybird were removed, dates were replaced and making all lines uniform.
Vertical bars were the standard error within each treatment. Some are too small to display in the figure.
Point 12: Are you sure that this experimented scribed in Figure 3 is open field?
Response 12: We revised the methodology according to comments at No. 7 and No.4 from reviewer 2.
Response to reviewer 2 comment 4: In factor, we observed the population dynamics of cotton aphid for one month. After 10 days, the aphid size declined and predator feed less due to the reduced aphid abundance. After 20 days, the aphid population steadily dropped near to zero. Therefore, a total of 4 observations were used in our study (5, 10, 15, 20 days). In order to easily understand, were vised figure 1and 2 using 5, 10, 15 and 20 days instead of dates. However, because cotton aphids developed their abundance rapidly but reduced in 20-30 days, and then declined in the field, we concerned the effectiveness of biocontrol within 10 days in figure3. In Materials and Methods, we added the detail information (Line121 and Line 148-150).
Point 13: Did EBC declined or number of aphids? If EBC declined after release of predators then something is wrong. Please reconsider and rewrite this paragraph.
Response 13: Thank for very positive comments. We described incorrectly this result in Figure 3. According to the result, after the release of predators, the number of aphids per plant decreased rapidly, showing the high biocontrol efficiency. We have rewritten the results shown in Figure 3(Line 183-192). The specific result as follows:
In caged experiment in garden, the number of predators and initial number of aphids were the key factors determining the EBC. After releasing the predator(s), the number of aphidsof each treatment had declined rapidly both at 5 days (Predator: F=230.50, df=2, p<0.0001; Initial number of aphids: F=105.14, df=2, p<0.0001) and 10 days (Predator: F=92.88, df=2, p<0.0001; Initial number of aphids, F=94.14, df=2, p<0.0001). The effectiveness of biocontrol (EBC) was greater than 80% except for one treatment: the EBC of the treatment with the combination of an initial number of aphids of 800 and one released predator was significantly lower than of other treatments (Figure 3).
Point 14: This statement “Regardless of the initial number of aphids and the number of released predators, the effectiveness of bio-control (EBC) was greater than 80% after 5 and 10 days” is in direct conflict with previously highlighted statement.
Response 14: Agreed, according to our incorrectly described results, this statement is in direct conflict with previous statement. However, in fact, this statement is corresponding with the right results. We have described the correct results in details in response to comment 13 raised by notable reviewer 1
Point 15: ECB should be EBC
Responses 15: Revised as suggested (Line 203).
Point 16: Please explain what are the numbers above bars in figure3?
Response 16: We added this information in the legend of figure 3 (Line 212-214). The difference letters showed the mean of EBC that was different among each treatment in (a) and (b), one way ANOVA was used for comparison. Similarly, we processed same questions in figure 4 (Line 218-220).
Point 17: Cages? You stated several times that you perform open field experiments. If it is open field then there should be no cages. Please provide detailed Methodology in Mat and Met section about field experiments. Did you use isolation cages and what dimensions there were? What was used inside the isloation cages etc?
Response 17: Please see reply to your questions 7 where we explained this in the revised version of manuscript.
Point 18: There is no a) and b) on the Figure 5. Remove this from legend
Response 18: Deleted.
Point 19: Reconsider are those an open field experiments?
Response 19: We described more details on this experiment in reply to question 7 where we called it as “caged experiment in garden”.
Point 20: “Practically, it is hard to understand the biocontrol efficiency of a natural enemy without controlled experiments in an open field natural environment. Exclusion cages can be employed to estimate the EBC of natural enemies in the field, but it is time-consuming and laborious in larger areas. PGR is easier to calculate using the initial density, final density, and time interval (days)”
Response 20: Agreed and we fixed our description on Experiment 2.
Point 21: “Predator guilds” There are also parasitoids and pathogens which could and are important in cotton aphid regulation. If you would like to go further there is also significant influence of endosymbionts.
Response 21: Thank you for deep insight. We revised term “predator guilds” with “guild of nature enemies” that is includes predators, parasitoids and pathogens (Line 271).
Point 22: On line 289,“ratio should be 728 because a ratio less than this threshold should lead to successful biocontrol of aphid” Your modeling did not take into consideration all other factors mentioned in previous paragraph. Please clearly state limitations of your study and suggestion.
Response 22: Thank for valuable comments. We discussed limitations of our study, and explained the future work in this field. We added this in MS as “In our experiments as well as modeling, we merely considered the limited factors and simplified to estimate the threshold of prey/predator in effective control of cotton aphids. This experiments and modeling should be developed as multiple parameters to consider the practical threshold of prey/predator against cotton aphid with eco-friendly actions.” (Line 288-291).
Point 23: I cannot see relevance of this statement “Moreover, planting Bt cotton over large areas in China can conserve the natural enemies in cotton fields because of reducing the numbers and times of insecticide sprays” Please rewrite this paragraph to present possibilities of different management practices as well as potential of their combination with your results.
Response 23: Revised as “Moreover, the landscape of cropping had more diversity, and offers the higher percentage of nature enemies in cropping landscapes [50,51]. The soft insecticides were prevailed in agriculture production in China[52] , and the amount of insecticide also was declined rapidly with China regulation policy [53]” (Line 313-318).
Special thanks to you for your good comments and effort.

Reviewer 2 Report
Dear authors, your paper resulted formally correct, but I found some lacks in the material and method chapter, You wrote you monitored the completion of three generations of Aphid before to carry out the experiment. What really do you mean for generation in a species, which under lab conditions perform parthenogenetic anholocyclic life cycle? Which methodology do you have adopted to count the huge number of aphids you utilized in all the experiment? I suppose you adopted an estimation method and that would be very interesing to describe and to report in the papaer. Even some photos could be useful to show the infestation of the plants and the distribution of aphids on their surface.
Fhurter, in my opinion, you should cite in the discussion chapter, some methods to estimate the aphids population and their predators in open fields. To decribe the sampling methodologies more suited to make your data really useful for open field observations.
I have put my observation in comments in the pdf file.

Author Response
Dear reviewer:
Thanks very much for taking your time to review this manuscript. We really appreciate all your professional comments and suggestions. Those comments are all valuable and very helpful for revising and improving our paper, as well as the important guiding significance to our researches. We have studied comments carefully and have made correction which we hope meet with approval. Revised portion are marked in yellow in the manuscript. Please find our revisions in the re-submitted files. The main corrections in the paper and the responds to your comments are as flowing:
Point 1: How did you controlled the completion of three generations in a parthenogenetic species? In lab conditions we suppose that A. gossypii perform an anholocyclic life cycle.
Response 1: Thank you for your valuable comments. The life cycle of cotton aphid (A. gossypii) is an anholocyclic under laboratory conditions. For this, before the experiment, we cultured a sufficient number of cotton aphids in the greenhouse. After most aphids completed their first generation of reproduction, we removed the previous generation of aphids and continued until 3rd generation is completed.
Point 2: How many plants were grown in an enclosed insect-proof greenhouse of?
Response 2: Before the experiment, more about 2400 plants (one plant per pot) were prepared and cultivated in greenhouse, which was enough to meet the needs of the experiment (wee revised in Line: 102).
Point 3: Five replicates and all these treatments produce a very huge number of aphids. I calculated 84,000 aphids only for the experiment 1 and other 28,000 for exp. 2. A total of 112.000 aphids. Maybe the authors C.W.D. and Z.P., who performed investigation, adopted a method for extimate the number of aphids. Please describe that. Have they really counted all the specimens?Than they repeated all counts after 5 and 10 days. It is a really strong effort.
Response 3: Thank for your highly valuable concern. I can say your calculation is right. In our case, we have a team of 8 underground students to help us in these data collection. After 5 days, the work load was much less than the first time (although counting is really hard work, agreed).
Point 4: “Aphids on each plant were counted at 5 day intervals” How may observations?
Response 4: In factor, we observed the population dynamics of cotton aphid for one month. After 10 days, the aphid size declined and predator feed less due to the reduced aphid abundance. After 20 days, the aphid population steadily dropped near to zero. Therefore, a total of 4 observations were used in our study (5, 10, 15, 20 days). In order to easily understand, were vised figure 1and 2 using 5, 10, 15 and 20 days instead of dates. However, because cotton aphids developed their abundance rapidly but reduced in 20-30 days, and then declined in the field, we concerned the effectiveness of biocontrol within 10 days. In Materials and Methods, we added the detail information (Line 119 and Line 144-146).
Point 5: Please, introduce at this point the acronym PGR
Response 5: We add the PGR at the end of the population growth rate for clear understanding (Line 134).
Point 6: delete, it is a repetition
Response 6: Thank you so much for your careful check. We have deleted one duplicate formula.
Point 7: In this formula only ln(N0) resulted fracted by time. It is this the right formula? Could be this the right one?:r = (ln Nt - ln N0) / t
Response 7: In fact, it is same equation. We ensure this is the right formula. N0 is the initial number of aphids in each treatment (200,400 and 800). The value does not change with time. Nt was fracted by time. The value of Nt is the final number of aphids after 5 and 10 days of release of predators. The equation form which you mentioned is easier. So we accepted your suggest in here.
Point 8: “and the temperature was at a constant 24℃ which is 232the optimal temperature for aphid growth” this sentence does not deal with figure 3
Response 8: We agreed with you. The constant temperature(24, 27, 30℃), density and size of predator were all considered in figure 4, but density and size of predator were considered in figure 3 which was conducted in the cage in open garden-temperature is fluctuant from 9 to 35 ℃. We rewritten this paragraph (Line 254-263) as following:
Our laboratory experiment and caged experiment in garden (CEG) showed that ladybird beetle predators suppressed the numbers of cotton aphids in most cases but failed when the initial number of cotton aphids was 800 with one predator (Figure 3, 4). This means that one predator cannot suppress the population growth of aphids under these conditions due to the greater number of offspring. Especially, after ten days, when the temperature was at a constant 24℃ which is the optimal temperature for aphid growth (Figure 4). In contrast, the extreme temperature of 30℃ tended to in-crease the EBC in our study in treatments with a lower prey/predator ratio in the laboratory (Figure 4).
We tried our best to improve the manuscript and hope that the correction will meet with approval.
We appreciate for your warm work earnestly and good comments and suggestions.

Round 2
Reviewer 1 Report
The authors addressed all my coments and made significant improvment of the manuscript which is now ready for publication.
Best regards
Author Response
Dear Editor Vostic and Referees:Thank you for considering my manuscript for publication in Insects after minor changes. We are grateful to you and the reviewers for the valuable suggestions provided. Here are responses to the comments:
Comment 1: The comma has been replaced with a colon on line 159 and 174.
Comment 2: Duplicate point are deleted on line 315.
Comment 3: the denominator d.f. are added on line 187-189 and line 200-203.
Revised portion are marked in yellow in the manuscript.
Thank you and best regards.
Yours Sincerely,
Zhang Ping
Reviewer 2 Report
Dear authors I have just reported two minor, formal corrections with my comments in the MS.

Author Response
Dear Referee:We are grateful to you for the valuable suggestions provided. Here are responses to the comments:
Comment 1: The comma has been replaced with a colon on line 159 and 174.
Comment 2: Duplicate point are deleted on line 315.
Comment 3: the denominator d.f. are added on line 187-189 and line 200-203.
Revised portion are marked in yellow in the manuscript.
Thank you and best regards.
Yours Sincerely,
Zhang Ping

This manuscript is a resubmission of an earlier submission. The following is a list of the peer review reports and author responses from that submission.
Round 1
Reviewer 1 Report
The authors described a study of the prey and predator ratio for effective biocontrol of cotton aphid in the seedling stage. Although the authors tried to give some new information, the present condition of the manuscript is not suitable for publication. There are a number of issues I have with their materials and results sections.
To begin with, the whole document included several spacing and grammatical errors. There are a few phrases that aren't as fluid as others. As a result, it makes the reviewer's job more difficult. The reviewer attempts to assist the authors in improving the article in the attached document.
Second, there is a lack of clarity in the materials and methods parts. Ladybird beetles were utilized to control aphid numbers in their research. The primary worry is how they contaminated each plant with precisely 200, 400, and 800 aphids. It's also unclear how many beetles per plant sample were dispersed.
The way the population growth rate of cotton aphids was described did not satisfy the reviewer. Why does population growth accelerate with time? How about a single predator's control rate? Why was the efficacy low when the aphid population reached 800?
How do the authors compare the mean differences between the aphid densities in Figure 3? What type of investigation did they do on that specific figure? What distinguishes figures 3 and 4?
No statistical analysis was carried out in Figure 4. When cotton aphids numbered 800, why was EBC negative (-) against them?
The discussion part is also not clear. Overall, I propose rejecting the paper owing to its lack of concentration. In addition, the manuscript file including revision was attached.
